# Evaluation of the Potential of Lipid-Extracted *Chlorella vulgaris* Residue for *Yarrowia lipolytica* Growth at Different pH Levels

**DOI:** 10.3390/md20040264

**Published:** 2022-04-13

**Authors:** Guillaume Delfau-Bonnet, Nabila Imatoukene, Tiphaine Clément, Michel Lopez, Florent Allais, Anne-Lise Hantson

**Affiliations:** 1Chemical and Biochemical Process Engineering Unit, Faculty of Engineering, University of Mons, 7000 Mons, Belgium; delfaug@yahoo.com; 2Unite Recherche et Developpement Agro-Biotechnologies Industrielles (URD ABI), Centre Europeen de Biotechnologie et Bieconomie (CEBB), AgroParisTech, 51110 Pomacle, France; nabila.imatoukene@agroparistech.fr (N.I.); tiphaine.clement@inrae.fr (T.C.); michel.lopez@agroparistech.fr (M.L.); florent.allais@agroparistech.fr (F.A.)

**Keywords:** *Yarrowia lipolytica*, *Chlorella vulgaris*, growth, fatty acids

## Abstract

Projections show that the cultivation of microalgae will extend to the production of bio-based compounds, such as biofuels, cosmetics, and medicines. This will generate co-products or residues that will need to be valorized to reduce the environmental impact and the cost of the process. This study explored the ability of lipid-extracted *Chlorella vulgaris* residue as a sole carbon and nitrogen source for growing oleaginous yeasts without any pretreatment. Both wild-type *Yarrowia lipolytica* W29 and mutant JMY3501 (which was designed to accumulate more lipids without their remobilization or degradation) showed a similar growth rate of 0.28 h^−1^ at different pH levels (3.5, 5.5, and 7.5). However, the W29 cell growth had the best cell number on microalgal residue at a pH of 7.5, while three times fewer cells were produced at all pH levels when JMY3501 was grown on microalgal residue. The JMY3501 growth curves were similar at pH 3.5, 5.5, and 7.5, while the fatty-acid composition differed significantly, with an accumulation of α-linolenic acid on microalgal residue at a pH of 7.5. Our results demonstrate the potential valorization of *Chlorella vulgaris* residue for *Yarrowia lipolytica* growth and the positive effect of a pH of 7.5 on the fatty acid profile.

## 1. Introduction

Microalgae are a diverse phylogenetic group of unicellular photosynthetic organisms. They represent species that live in a wide range of fresh and marine environments. Due to their diversity and ability to capture and convert carbon dioxide, they are promising candidates for the production of bio-based building blocks and molecules. These molecules can be used in animal feed, food supplements, nutraceuticals, cosmetics, and pharmaceuticals [1,2,3,4]. Microalgae have also been proposed as a promising candidate for lipid accumulation due to their adaptability to environmental stresses and rapid growth rate, as well as the large number of biochemical compounds they produce [5,6,7]. Among microalgae, three main genera have been widely used: *Chlamydomonas* sp., *Scenedesmus* sp., and *Chlorella* sp. [8]. Increased use of *Chlorella vulgaris* to produce polyunsaturated fatty acids (PUFA) has raised concerns regarding the environmental impact of its by-products. Indeed, delipidated algal co-products account for approximately 65% of the harvested biomass. It has been reported that the global production of microalgae was 15,000 tons/year [9]. In 2020, the production of *Chlorella vulgaris* within European Union was 82 tons [10]. However, there is lack of studies regarding effective utilization of the microalgal residues, which is left after lipid extraction but still enriched with significant amount of proteins, carbohydrates, or cellulose, so we aimed to study *Chlorella vulgaris* residue as substrate [11].

To decrease the quantity of by-products, residual microalgae biomass obtained after lipids extraction has been considered as a promising substrate for oleaginous yeasts, as residual microalgae biomass is a renewable source of high levels of organic carbon and nitrogen. Many studies on growth and lipid production from oleaginous microorganisms have used residual microalgae biomass after lipids extraction as the sole carbon and nitrogen source [12,13]. However, delipidated microalgae are not usable as a substrate because the remaining components of the cell wall and membrane are not bioavailable for microorganisms [14]. Due to the recalcitrance of algal by-products, pretreatments have been applied to lipid-extracted microalgae before their use [4,15]. As an example of a pretreatment, chemical hydrolysis often generates inhibitory substances that can hinder or abolish the growth of microorganisms cultivated in the resulting hydrolysates [12,16,17,18]. Seo et al. used physical means, such as ultra-sonication or hydrothermal cavitation, to solubilize residual algal biomass from biodiesel production; however, high energy consumption and scalability are not easy issues to solve [13]. Some researchers reported the use of *C. vulgaris* hydrolysates generated after lipids extraction and enzymatic treatment for the production of bioethanol with a wide diversity of microorganisms, including *E. coli*, *P. stipitis*, and *S. cerevisiae* [19,20,21,22]. Moreover, other researchers engineered a strand of *E. coli* that transformed *Chlorella emersonii* hydrolysate to 2-pyrone 4,6-dicarboxylic acid [23]. Shahi et al. attempted to produce bio-oil from a residual biomass of microalgae *Dunaliella sp*. after lipids extraction using hydrothermal liquefaction and achieved a bio-oil yield of 11.8% in dry matter [17].

To simplify the fermentation process and avoid the residual toxic compounds resulting from pretreatment, this study did not submit the *C. vulgaris* residue substrate to any mechanical, chemical, or enzymatic hydrolysis to enhance assimilation of nutrients by oleaginous yeasts. 

*Y. lipolytica* is one of the oleaginous yeasts that can accumulate more than 30% of its dry weight in lipids [24]. It represents an attractive source of valuable compounds that can be used in biotechnology, pharmaceuticals, and the food industry [25,26]. *Y. lipolytica* can grow in a wide range of pH levels and temperatures and can metabolize many substrates, including alkane derivatives and hydrophobic substrates, such as lipids [27,28]. 

Additionally, *Y. lipolytica* can tolerate some organic solvents, such as those used for lipids extraction, such as hexane or chloroform/methanol [29]. Carbon and nitrogen sources, C/N ratio, temperature, dissolved oxygen, agitation, and pH have been shown to influence the quantity and quality of lipids accumulation significantly, particularly the saturated to unsaturated fatty acids ratio [30,31,32,33]. Most reported experiments have used wild-type *Y. lipolytica* strains, such as W29. However, mutant *Y. lipolytica* strains, such as JMY3501, were designed to accumulate more than 45% lipids through the deletion of genes involved in lipids catabolism and triacylglycerol (TAG) intracellular remobilization [34]. The fatty acid profile measured in mutant yeast was compared with that synthesized on the W29 strain, both cultivated on microalgal substrate. 

In this study, we examined the potential use of lipid-extracted *C. vulgaris* residue as a sole carbon and nitrogen source for growing two strains of *Y. lipolytica* (W29 and JMY3501). The main objective of this study was to evaluate the impact of this original substrate on the growth rate and fatty acid profile. In addition to sterilization, the microalgal substrate did not undergo any physical, enzymatic, or chemical pretreatment, not only to prevent the generation of inhibitors but also to develop an eco-friendly fermentation process. Moreover, to achieve complete exploitation of microalgae biomass in a microalgae biorefinery, no carbon source was added. The effect of pH on both cell growth and the fatty acid profile was the major criterion considered in this work. 

## 2. Results

This study aimed to evaluate the possibility of using lipid-extracted *C. vulgaris* without pretreatment as a sole carbon and nitrogen source for the growth and fatty acid profile of *Y. lipolytica* strains. The study also assessed the impact of different pH levels on the growth and fatty acid profiles of wild-type W29 and mutant JMY3501 yeasts cultivated in *C. vulgaris* residue or on yeast-extract peptone dextrose (YPD) media. We used different studies to determine the pH levels to be tested. Although the best conditions for the growth of *Y. lipolytica* W29 have been reported as a pH of 5.5 and temperature of 28 °C [35], Zhang et al. concluded that the best pH for producing lipids was 2.0 [33]. Neutral and alkaline conditions have also been shown to enhance protein solubilization and dispersion of the microalgal cell wall [36]. Therefore, we performed the tests in this study at three pH levels: (i) a pH of 3.5 allowed for a better production of lipids, (ii) a pH of 5.5 provided better growth conditions for the yeasts, and (iii) a pH of 7.5 showed better dispersion of the substrate and therefore better accessibility. 

### 2.1. Biochemical and Elemental Analysis of C. vulgaris Residue 

In this study, we first determined the biomass composition of *C. vulgaris* after lipids extraction (Table 1). As expected, the main constituent in dry *C. vulgaris* residue was proteins at 40.1 ± 5.3% (*w/w* of dry biomass). Other components included residual fatty acids and ash, which were measured as 0.34 ± 0.02% and 11.2 ± 0.1% (*w/w* of dry biomass), respectively. Reducing carbohydrates composed about 15.6 ± 2.1% of the dry biomass.

The green algal lipid-extracted biomass showed high amounts of total carbon and nitrogen, representing 34.9 ± 0.4% and 7.8 ± 0.2%, respectively (Table 1). The acquired biomass data thus suggested that *C. vulgaris* could serve as a sole substrate source of carbon and nitrogen for microbial cultivation. It had an average carbon/nitrogen (C/N ratio) of 4.5, close to that of the YPD medium (C/N = 5.3) used as a control in this study. 

### 2.2. Growth of Wild-Type and Mutant Y. lipolytica Strains in Lipid-Extracted C. vulgaris Residue and Control YPD Media

We conducted all yeast cultures with the same batch of lipid-extracted *C. vulgaris* residue for 50 h to limit the impact of microalgal biochemical composition. Additionally, we performed all cultures in duplicate. Then, we evaluated the effect of pH on the cell growth of *Y. lipolytica* W29 and JMY3501 cultivated in YPD and *C. vulgaris* residue media.

#### 2.2.1. The Effect of pH on the Cell Growth of *Y. lipolytica* W29

Figure 1a compares the growth profiles of *Y. lipolytica* W29 in YPD medium at a pH of 5.5 and in lipid-extracted *C. vulgaris* residue medium at different pH levels. Table 2 shows the main parameters used to assess the growth of wild-type W29 and JMY3501 yeasts, in both the control YPD and *C. vulgaris* residue media. 

The specific growth rate of the wild-type strain W29 in the YPD and *C. vulgaris* residue media appeared similar regardless of pH level (3.5, 5.5, and 7.5). We obtained a maximum specific growth rate (µ_max_) of 0.28 h^−1^ (Table 2). 

However, the wild-type W29 cell concentration showed major differences depending on the culture conditions. When the wild-type strain was cultivated on the microalgal residue medium at a pH of 7.5, the stationary phase was reached after 25 h. In other pH levels, the stationary phase was reached after only 15 h (Table 2, Figure 1a). The final cell concentration of the W29 strain in the *C. vulgaris* medium at a pH of 5.5 or 7.5 was above 70% of the final cell concentration at a pH of 3.5 (Table 2). 

#### 2.2.2. The Effect of pH on the Cell Growth of JMY3501 

We then examined the effect of pH (3.5, 5.5, and 7.5) on the growth of the JMY3501 strain in the *C. vulgaris* residue medium compared with that at a pH of 5.5 in the YPD medium. JMY3501 is a genetically modified strain in which the β-oxidation pathway, intracellular lipid mobilization, and alkaline protease have been blocked, rendering overexpression of the genes that push and pull TAG biosynthesis [34]. We compared the results with those of the W29 strain cultivated in the same media. The mutant strain showed almost equivalent microbial growth compared to the W29 strain (0.28 ± 0.01 h^−1^, Table 2) at different pH levels when the same residue was used (Figure 1b, Table 2). However, we obtained a maximum specific growth rate (µ_max_) of 0.3238 ± 0.0004 h^−1^ when we cultivated the JMY3501 strain in the YPD medium at a pH of 5.5 (Table 2). We did not observe a significant difference between the two strains under the same conditions in terms of maximum specific growth rates, nor did we observe a lag time, regardless of the pH level or medium used (*C. vulgaris* residue and YPD at pH 5.5). The cell concentration of the JMY3501 strain was similar for the three pH levels, ranging from 62.8 ± 18.3 to 77.1 ± 13.7 10^6^ cells.ml^−1^ after 50 h of cultivation (Table 2). Thus, the growth of JMY3501 cultivated in the *C. vulgaris* residue medium did not seem to be influenced by the pH variation in the medium. 

These results clearly show that the growth profile of the W29 and JMY3501 strains differed in YPD and *C. vulgaris* residue media. The wild-type strain had more difficulty exploiting the microalgal substrate at a pH of 3.5. Although its cell growth was three times higher at pH levels of 5.5 and 7.5 in *C. vulgaris* residue, this growth was still 4–5 times lower compared with that in the YPD medium. Regarding cultures of mutant yeast in *C. vulgaris* residue, performance of W29 was improved by 30% regardless of pH compared with the JMY3501 yeast at pH levels of 5.5 and 7.5 in the microalgal residue used as the sole source of carbon and nitrogen. Though we did not observe changes in growth rate analysis at different pH levels, we expected to see changes in the fatty acids profile.

### 2.3. Fatty Acid Profiles of Wild-Type and Mutant Y. lipolytica Strains 

Several factors, such as the physiological state, dissolved oxygen, agitation, osmotic pressure, nutrients, and carbon and nitrogen sources, influence the biosynthesis of fatty acid methyl esters (FAMEs). In this study, we measured the effect of the substrate and pH levels on the fatty acid profiles of the yeast strains.

#### 2.3.1. Effect of Different pH Levels on the Fatty Acid Profile of *Y. lipolytica* W29 

We compared the fatty acid profiles of the W29 strain after 50 h of cultivation in both media: YPD at a pH of 5.5 (yeast control) and *C. vulgaris* residue at different pH levels. We incubated blank *C. vulgaris* residue at a pH of 7.5 for 50 h to evaluate the temporal evolution of fatty acids without yeast growth. We selected this pH because of the improved substrate homogenization. Fatty acid concentrations were quantified using a gas chromatography–flame ionization detector (GC-FID). The results are shown in Figure 2a,b. 

In the YPD medium, oleic acid (C18:1) was the predominant fatty acid found in the wild-type W29 strain (12.8 mg·g^−1^ of biomass dry weight), while linolenic acid (C18:2) reached 7.2 mg.g^−1^ of the dry biomass. Palmitic acid (C16:0) and palmitoleic acid (C16:1) were identified, but no linolenic acid (C18:3) was detected in the W29 strain cultivated in the YPD medium. Regardless of pH, the wild-type strain fermented in the *C. vulgaris* residue medium showed no significant changes regarding the fatty acid profile present in the initial algal raw material, as shown in Figure 2a. 

FAMEs analyses were performed on the biomass produced after 50 h of yeast growth without any separation of the yeasts and remaining *C. vulgaris* residue. To demonstrate a potential biosynthesis of FAMEs by the wild-type W29, we subtracted the fatty acid quantities present in the initial *C. vulgaris* residue from the final fatty acid content measured in the recovered biomass. Thus, the fatty acid content shown in Figure 2b and Figure 3b was likely underestimated.

As shown in Figure 2b, the wild-type strain catabolized C16:0 and C18:3 at different pH levels (3.5, 5.5, and 7.5; Kruskal–Wallis test, *p*-value < 0.05), while C16:1, C18:1, and C18:2 seemed to be anabolized by the *Y. lipolytica* W29 at pH levels of 3.5, 5.5, and 7.5 (Kruskal–Wallis test, *p*-value < 0.02). As seen in Table 3, the total quantity of FAMEs and biomass remain unchanged in the wild-type strain after 50 h of cultivation in the *C. vulgaris* residue medium (regardless of pH).

#### 2.3.2. Effect of Different pH Levels on the Fatty Acid Profile of *Y. lipolytica* JMY3501 

When comparing the wild-type and modified yeast strains cultivated on the YPD medium (Figure 2a and Figure 3a), we noticed similar fatty acid profiles. In contrast, the JMY3501 strain was genetically engineered to be incapable of degrading fatty acids and remobilizing TAGs. On the YPD medium, the average fatty acid accumulation in the JMY3501 strain (26.5 ± 6.1 mg.g^−1^) was equivalent to that of the W29 strain (25.4 ± 18.4 mg.g^−1^). This showed the importance of a high C/N ratio for fatty acid accumulation [37]. 

The fatty acid content of the JMY3501 strain cultivated in the *C. vulgaris* residue medium at different pH levels indicated the presence mainly of long-chain fatty acids with 16 and 18 carbon atoms. To obtain an accurate understanding of the fatty acid metabolism of the mutant yeast, we calculated the final quantity of each identified fatty acid after deduction of the FAMEs quantity present in the microalgal raw material control bioreactor (Figure 3b). In addition to the fatty acid profile of the *C. vulgaris* residue, JMY3501 cultivation appears to induce an increase in C18:2 synthesis regardless of pH (Kruskal–Wallis test, *p*-value < 0.02), while C18:1 rose only at pH levels of 3.5 and 7.5 (Kruskal–Wallis test, *p*-value < 0.02). 

JMY3501 growth at a pH of 7.5 in *C. vulgaris* appeared to be the only culture showing an increase in the total FAMEs associated with an unchanged quantity of biomass after 50 h of cultivation (Table 3). Under these specific conditions, fatty acid quantities differed from the other tested conditions. Indeed, at a pH of 7.5, the JMY3501 strain cultivated on *C. vulgaris* residue synthesized significant quantities of C16:0, C18:2, and C18:3 (Kruskal–Wallis test, *p*-value < 0.02; Figure 3b).

## 3. Discussion

The broad objective of this study was to assess the potential use of lipid-extracted *C. vulgaris* residue as a sole source of carbon and nitrogen without pretreatment for growing two strains of *Y. lipolytica* (W29 and JMY3501). We examined the effect of substrate and pH on both cell growth and the fatty acid profile. 

First, we compared the cell growth of wild-type and mutant *Y. lipolytica* strains on lipid-extracted *C. vulgaris* residue and control YPD media. Different pH levels (3.5, 5.5, and 7.5) were evaluated. The specific growth rates of the wild-type strain W29 in the YPD and *C. vulgaris* residue media were similar regardless of pH (3.5, 5.5, and 7.5). A maximum specific growth rate (µ_max_) of 0.28 h^−1^ was obtained. This finding matches those reported in studies in which *Y. lipolytica* W29 was cultivated in glucose [37,38,39]. However, yeast cellular growth (cell counts) was directly impacted by the relevant strain/substrate combination, especially with complex raw material. This explains the differences observed regarding the maximal growth of the W29 and JMY3501 strains cultivated on *C. vulgaris* residue.

The final cell concentration of the W29 strain in the *C. vulgaris* medium at a pH of 5.5 or 7.5 was above 70% of the final cell concentration at a pH of 3.5 (Table 2). This discrepancy may have been due to the reduced accessibility of the nutrients contained in *C. vulgaris* residue because of the pH and lack of pretreatments. Since proteins and polysaccharides are major microalgal macroconstituents, their metabolization requires the secretion of hydrolytic enzymes, such as proteases, from the wild-type W29 strain. This strain had the capacity to secrete alkaline proteases at a pH of 7.5; the lack of alkaline protease secretion by the JMY3501 strain prevented protein degradation and thus the release of amino acids and peptides from the microalgal substrate. Therefore, cell growth in the microalgal medium appeared to be three times lower in the deleted *xpr2* gene JMY3501 mutant at pH levels of 5.5 and 7.5 than in the wild-type W29 strain at the same pH levels [40].

To simplify the process and avoid the residual toxic compounds that result from pretreatment, we did not submit the *C. vulgaris* residue substrate to any mechanical, chemical, or enzymatic hydrolysis to enhance the yeasts’ assimilation of nutrients [19]. Therefore, both *Y. lipolytica* strains had less readily available carbon and nitrogen sources for their growth. Several studies have explored the feasibility of using microalgae hydrolysate for microbial growth [12,13,17]. However, complex detoxification would be necessary prior to fermentation; hence, these researchers did not undertake the additional costs and eco-friendly aspects that were adopted in our study. 

Regarding the lipids accumulation within the JMY3501 cultivated on *C. vulgaris* residue at different pH levels, the cultivation of JMY3501 on microalgal residue with a C/N ratio of 4.5 achieved a moderate cellular lipid content (5.3 ± 0.6 mg.g^−1^ of biomass dry weight) at a pH of 7.5, similar to that achieved by the W29 strain under the same conditions (5.1 ± 0.7 mg.g^−1^ of biomass dry weight). A pH of 7.5 appeared to have a positive effect on lipid accumulation when the mutant strain was cultivated on microalgal residue as its sole source of nutrients (27.6 ± 1.2 mg of total FAMEs in the JMY3501 biomass vs. 17.8 ± 4.1 mg in the *C. vulgaris* residue). The overall production efficiency of FAMEs was similar for the two strains (W29 and JMY3501). However, the cell growth of the JMY3501 strain was weaker. Therefore, the conversion yield of *C. vulgaris* residue into fatty acids was better in the JMY3501 strain. Previous research has shown that lipid accumulation is highly dependent on the C/N ratio and is induced by nitrogen starvation [24,41] and has reported that the optimal C/N ratio for lipids accumulation is around 30 [42,43]. *C. vulgaris* residue has a high nitrogen content and a low C/N ratio, which is not favorable for lipid production based on the nitrogen-limitation strategy. In this study, the JMY3501 strain deleted in the alkaline protease was chosen to promote lipid synthesis requiring a high C/N ratio. This choice was made at the expense of cell cultures requiring metabolizable nitrogen to direct lipid anabolism.

The FAMEs profiles and mass balances of the W29 and JMY3501 strains showed that pH and culture media influence the balance between lipid anabolism and catabolism. In both yeasts, lipid synthesis tended to increase when the pH in the microalgal residue media increased. This phenomenon was even more pronounced in the JMY3501 strain, with significant metabolization of C18:3 at a pH of 3.5; lipid profile changes were limited at a pH of 5.5, while a clear trend toward lipid anabolism occurred at a pH of 7.5 with the noteworthy appearance of C18:3. The synthesis of α-linolenic acid (C18:3 [c9, c12, c15]) in the JMY3501 strain cultivated on *C. vulgaris* residue after 50 h (Figure 3b) was confirmed by GC-MS (data not shown). *Y. lipolytica* is known not to synthesize this fatty acid naturally [44,45,46,47]. 

When the JMY3501 strain was cultivated on *C. vulgaris* residue, the amount of fatty acids varied depending on the pH level (Figure 3a). The best production of C18:1 and C18:2 was observed at a pH of 3.5, while production of C18:3 was higher at a pH of 7.5. Previous studies have found that when *Y. lipolytica* is cultivated in a canola oil substrate, about 22% of the total FAMEs are C18:3. This C18:3 is not synthesized but originates from the plant after being accumulated by the yeast [30,31]. Few researchers have reported significant quantities of C18:3 in different *Y. lipolytica* strains grown on glucose-based medium. In 2020, Carsanba et al. showed that C18:3 content in wild-type strains W29 and H917 reached 4.1% and 14.2%, respectively, based on the total lipids and 19.9% in the mutant strain Pol1dL. In 2014, Mattanna et al. described the *Y. lipolytica* strain QU22 isolated in artisanal Brazilian cheeses whose total lipid fraction contained 13% of C18:3 [44,48]. However, no information was provided about the C18:3 synthetic pathways in *Y. lipolytica* strains. In oilseeds and microorganisms, the synthesis of α-linolenic acid occurs through desaturation of linoleic acid (C18:2 c9, c12) with Δ15 desaturase. Synthesis can also be carried out by a Δ12-15 desaturase in several species of fungi [45,49,50]. 

In this study, no C18:3 was measured in the wild-type W29 or JMY3501 strain cultivated in YPD medium. This PUFA was only synthesized when the mutant JMY3501 was grown at a pH of 7.5 on *C. vulgaris* residue (Kruskal–Wallis test, *p*-value = 0.02). This C18:3 synthesis could be attributed to an unknown factor present in *C. vulgaris* residue released at a pH of 7.5 that induced the synthesis in the JMY3501 strain specifically engineered to inhibit lipid catabolism. These results provide a basis for further exploration of C18:3 production by *Y. lipolytica*: genetic modifications can be combined with i) the overexpression of cellulases and proteases to release polysaccharides and proteins from microalgal residue; ii) addition of a carbon source in the medium, such as glycerol; and iii) chemical, enzymatic, or physical pretreatments of the residue. 

## 4. Materials and Methods

### 4.1. Strains and Plasmid Genotype 

*Y. lipolytica* strain W29 (CLIB89, *MATa* WT) was obtained from the Centre International de Ressources Microbiennes (CIRM), while the JMY3501 strain was provided by the laboratory Biologie Intégrative du Métabolisme Lipidique INRAE, UMR1319, Jouy-en-Josas, France. *Y. lipolytica* W29 is a wild-type strain [41]. The prototrophic strain JMY3501 (*MATa ura3-302 leu2-270 xpr2-322 Δpox1-6 Δtgl4 + pTEF-DGA2 LEU2ex + pTEF-GPD1 URA3ex*) was previously described by Lazar et al. and is unable to degrade fatty acids or remobilize TAG due to deletion of the POX genes and the TGL4 gene, respectively [34]. The deletion of POX1-6 completely blocks β-oxidation in the peroxisome [51], whereas deletion of TGL4 prevents the release of fatty acids from the lipid body [52]. Additionally, this strain overexpresses *YlDGA2* and *YlGPD1,* which push and pull TAG biosynthesis. GPD1 is involved in glycerol-3-phosphate formation, a precursor of TAG [53], while the DGA2 gene encodes the acyltransferase involved in the final step of TAG formation [54]. The *xpr2* gene, which directs the synthesis of a thermosensitive alkaline protease, was also deleted.

### 4.2. Culture Conditions

YPD medium was used to preculture yeasts at 28 °C and 150 rpm for 15 h. The medium contained 10 g.L^−1^ yeast extract (VWR Chemicals ®), 20 g.L^−1^ peptone (Universal Peptone M66, Merck®), and 20 g.L^−1^ glucose (VWR chemicals ®). The W29 and JMY3501 strains were cultivated in a 500 mL bioreactor (INFORS® HT, Switzerland) with a final operating volume of 300 mL at 28 °C. The culture media were inoculated at less than 1.5% (*v:v*) to obtain an optical density of 0.1, with cultures grown overnight in YPD in shake flasks. Two media were used: a YPD medium and a lipid-extracted *C. vulgaris* residue medium. The dissolved oxygen was set up at a 30% saturation level and was controlled by stirring and O_2_ injection. The working pH (adjusted after sterilization in an autoclave at 121 °C for 20 min) was maintained using KOH 4M or H_2_SO_4_ 2M. All experiment parameters were monitored using Eve® Infors software. A total of 0.5 mL antifoam (silicon anti-foam, Merck®, Germany) was added before *C. vulgaris* residue sterilization.

### 4.3. Fermentation of C. vulgaris by Y. lipolytica

This study was performed to assess *Y. lipolytica* growth on microalgae residue. *C. vulgaris sp.* provided by the Allmicroalgae® Company (Lisbon, Portugal) was employed as a feedstock for yeast growth and fatty acid production. Before *C. vulgaris* residue fermentation, the lipids content in the microalgae was removed by Soxhlet extraction [55]. Dry microalgae (10 g) was weighed into a cellulose thimble (25 × 100 mm, VWR®) and placed in a Soxhlet apparatus. Lipids were extracted using 150 mL of azeotropic chloroform/methanol mixture for 24 h (AnalaR NORMAPUR®, VWR chemicals®). After extraction, the solvents in the solid fraction (microalgae) were evaporated under a chemical hood for 15 h at ambient temperature (25 °C) to obtain a lipid-extracted *C. vulgaris* biomass.

Microalgal biomass (30 g.L^−1^ of *C. vulgaris*) residue after lipids extraction was used as a substrate for *Y. lipolytica* growth. Two control experiments using YPD medium and *C. vulgaris* residue without yeasts were performed under the same conditions.

The effect of pH (3.5, 5.5, and 7.5) on the growth and fatty acid profile of *Y. lipolytica* was studied. 

### 4.4. Analytical Procedure

#### 4.4.1. Determination of Yeast Growth

To determine the yeast cell density, 10 mL was taken 12 times and 50 mL was taken once (after 13 h of growth) from the bioreactor. On average, 10–15 mL of acid or base was added during the culture, so the final bioreactor volume was around 145 mL. 

The yeast cell density was quantified within a Bürker cell chamber. The growth parameters were calculated with Excel^®^ using the natural logarithm of cells.mL^−1^ versus time (h). The curve slope during the exponential phase provided the maximum growth rate (µ_max_) (Equation (1)). Exponential phase tangent intersection with the *x*-axis provided the lag time (λ). Generation time (Equation (2)) was calculated from the growth rate (Equation (1)). For a better growth curve representation, the data were transformed using the Gompertz model (Equation (3)) [56].
(1)$${µ}_{max}=\frac{dN}{dt}$$
(2)$$G=\frac{ln2}{{µ}_{max}}$$
(3)$$X\left(t\right)=ln\frac{N_{x_{max}}}{N_{0}}e^{-e^{\left(\frac{{µ}_{max}\;e}{\ln\frac{N_{x_{max}}}{N_{0}}}\;\left(\lambda -t\right)+1\right)}}$$

#### 4.4.2. Biochemical and Elementary Characterization of the Microalgae Strain after Lipid Extraction

The concentration of reducing carbohydrates was determined according to the modified Miller method [57]. Using 10 mL of H_2_SO_4_ 2M at 90 °C for 6 h in a reaction tube (Reacti-Vial^TM^, Thermoscientific®), 50 mg of freeze-dried *C. vulgaris* residue was hydrolyzed. A total of 2 mL of hydrolysate was transferred into new tubes and centrifuged with a MiniSpin microcentrifuge (Eppendorf®) at 5300 rpm for 5 min. The supernatant was filtered with a 0.45 µm PVDF syringe filter (VWR®), and 0.67 mL of 3,5-dinitrosalycilic acid solution 2.8M (NaOH 0.6M, KNaC_4_H_4_O_6_·4H_2_O 1.06M) was added, incubated for 5 min at 90 °C, and cooled for 5 min. The absorbance was measured at 530 nm (Hach® DR1900).

The protein content was determined as follows: after freeze-drying, 5 mg of *C. vulgaris* residue was placed into mill jars in the presence of 2 mL of Sodium Dodecyl Sulfate (SDS) solution 6.9 mM and ground for 30 min at 30 Hz using a Beater bead miller (Star-Beater, VWR®). The total proteins were estimated using the Pierce^TM^ BCA (BiCinchoninic Acid) Protein Assay kit from Thermoscientific^®^ (lot VI310505). The total nitrogen was assessed with the Kjeldahl method using 100 mg of lyophilized *C. vulgaris* residue as recommended by ISO 11,261 [58]. The total organic carbon was determined on 100 mg of dry *C. vulgaris* residue using an SSM-500A Solid Sample Module combined with a TOC-VCS/CP analyzer (Shimadzu®). 

The ash was determined using the NREL (National Renewable Energy Laboratory, Golden, Colorado) method [59]. This was performed on 500 mg of freeze-dried *C. vulgaris* residue for 4 h at 565 °C.

#### 4.4.3. Lipids Extraction

The lipids content was extracted from *Y. lipolytica* using a modified Bligh and Dyer method [60]. First, 50 mg of freeze-dried yeasts was placed in conical centrifuge tubes (15 mL, Pyrex™). A total of 7 mL of a chloroform/methanol (AnalaR, NORMAPUR®, VWR®) mixture (1:1, *v:v*) was added and homogenized for 30 s with a vortex. Then, 2 mL of demineralized water was added. The mixture was vigorously shaken for 15 min at 1800 rpm (IKA®-VIBRAX-VXR, Leuven, Belgium) and then centrifuged for 15 min at 3000 rpm (Labofuge Heraeux Sepatech®). The organic layer was collected in a new tube and kept at an ambient temperature. A second centrifugation was performed after 2 mL of chloroform/methanol (1:1; *v:v*; AnalaR, NORMAPUR®, VWR®) and 1 mL of NaCl 0.5M (to improve extraction yield) were added. Finally, 2 mL of chloroform was added, and a final centrifugation and collection of organic layers were performed. The organic phase (lower) was recovered and evaporated under a dry air flow at 50 °C. 

#### 4.4.4. FAMEs Derivatization

The extracted lipids were transmethylated as follows: 4 mL of methanolic HCl 3M (Sigma-Aldrich®, St. Quentin Fallavier Cedex, France) was added to the extracted lipids. The reaction was conducted at 50 °C for 6 h to allow the formation of FAMEs, which were then extracted by adding 6 mL of *n*-heptane (AnalaR, NORMAPUR®, VWR®). Samples were vortexed rigorously for 30 s and decanted for 5 min. A fraction of the upper phase was collected with a Pasteur pipette and transferred to a new tube. Internal standard (methyl nonanoate, Supelco, Sigma-Aldrich®) was added to the samples to achieve a final concentration of 100 ppm. 

#### 4.4.5. FAMEs Gas Chromatography Analysis

The FAMEs were analyzed using a Shimadzu® GC-2010 Plus Gas Chromatograph (GC) coupled with a flame ionization detector (FID) equipped with an Rt-2560 capillary column (100 m length, 0.25 mm diameter, 0.20 μm film thickness, Restek®). A modified AOAC method 996.06 was used. One-microliter samples of FAMEs were injected. The carrier gas was helium with a flow rate of 1.74 mL.min^−1^. The temperature program was as follows: after injection, the temperature was initially held at 100 °C for 4 min and then increased to 240 °C with a 3 °C.min^−1^ heating ramp. The injector temperature was set to 225 °C, and the FAMEs were detected with flame ionization using a detector set at 285 °C. The flow rate in the detector was a mix of three gases: 40 mL.min^−1^ of H_2_, 30 mL.min^−1^ of He, and 400 mL.min^−1^ of air. Standard molecules of the FAMEs mixture (C4–C24) were injected for FAME identification (Supelco® 37 Component Fatty Acid Methyl Ester Mix, Sigma-Aldrich®). 

## Figures and Tables

**Figure 1 marinedrugs-20-00264-f001:**
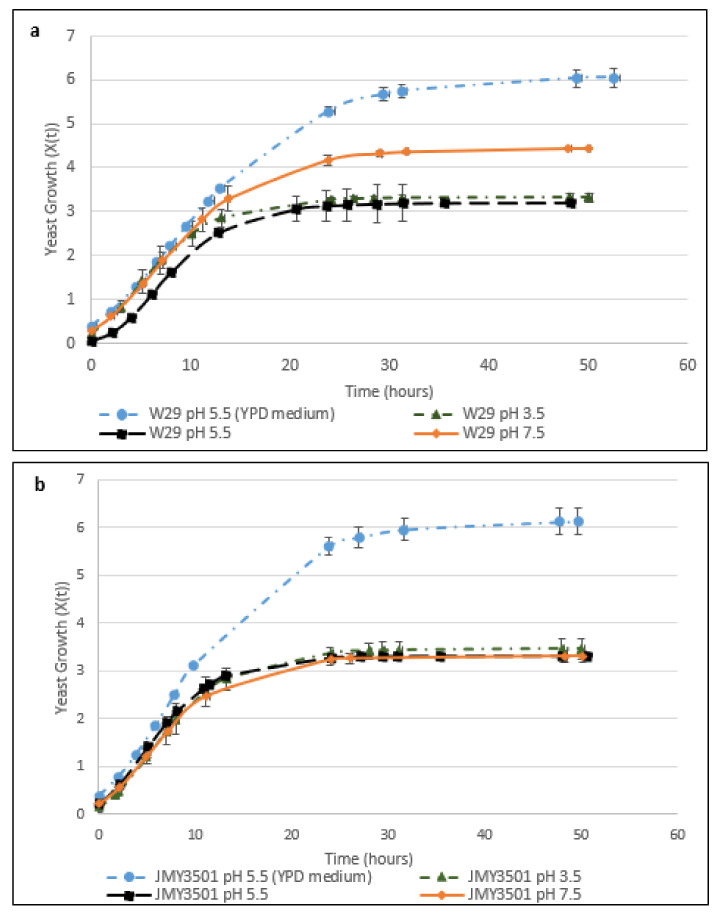
(**a**) Growth curves of *Y. lipolytica* W29 on YPD medium at a pH of 5.5 (blue circle) and on *C. vulgaris* residue at pH levels of 3.5 (green triangle), 5.5 (black square), and 7.5 (orange diamond). (**b**) Growth curves of *Y. lipolytica* JMY3501 on YPD medium at a pH of 5.5 (blue circle) and on *C. vulgaris* residue at pH levels of 3.5 (green triangle), 5.5 (black square), and 7.5 (orange diamond). Growth curves were obtained using the Gompertz model.

**Figure 2 marinedrugs-20-00264-f002:**
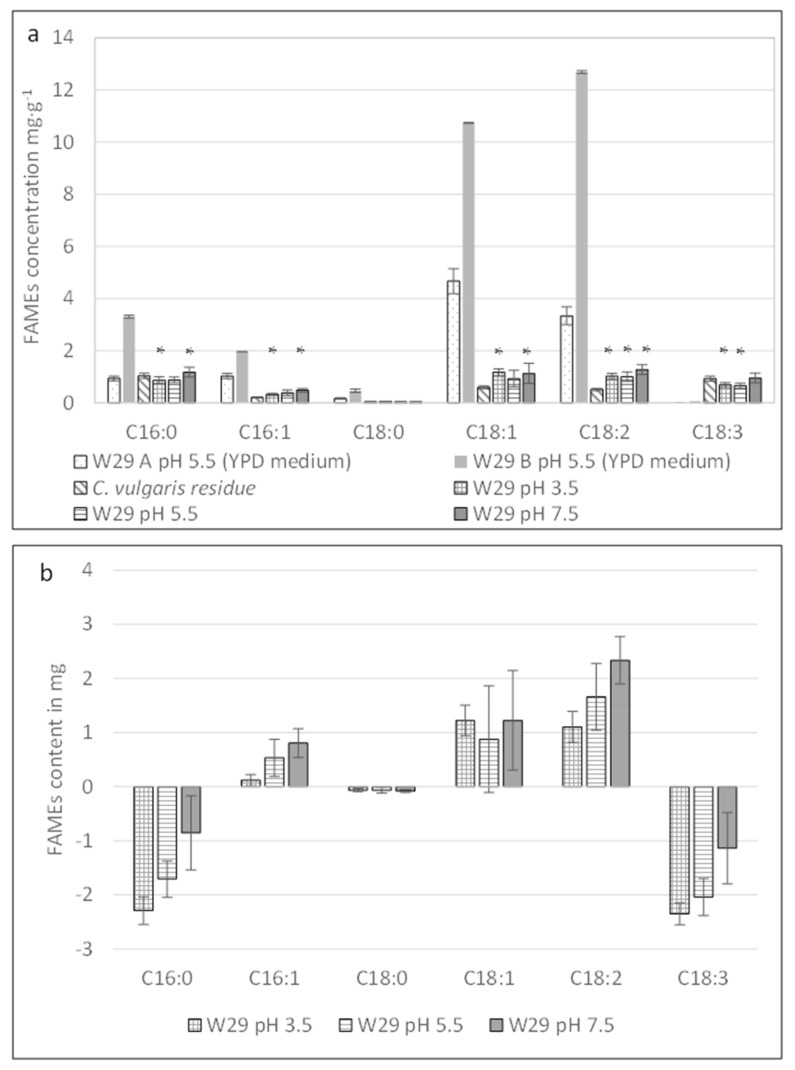
(**a**) Raw data regarding fatty acid content in *Y. lipolytica* W29 cells cultivated in the YPD medium at a pH of 5.5 (A and B represent two different fermenters in the YPD medium), algal raw material control (*C. vulgaris* residue incubated at a pH of 7.5), and the W29 strain cultivated in *C. vulgaris* residue for 50 h at different pH levels of 3.5, 5.5, and 7.5. (A statistical difference was found between the initial concentration in the *C. vulgaris* residue and final concentration of fatty acids after fermentation in the same residue; Kruskal–Wallis test, n = 4, ddl = 1, * significant at a *p*-value < 0.05.) (**b**) Evolution of fatty acid quantities in the fermenter after 50 h cultivation of the W29 strain in *C. vulgaris* residue. These values were obtained by subtraction of the initial fatty acid quantities present in the *C. vulgaris* residue from the final fatty acid quantities present in the fermenter. (C16:0: palmitic acid; C16:1: palmitoleic acid; C18:0: stearic acid; C18:1: oleic acid; C18:2: linoleic acid; C18:3: α-linolenic acid.)

**Figure 3 marinedrugs-20-00264-f003:**
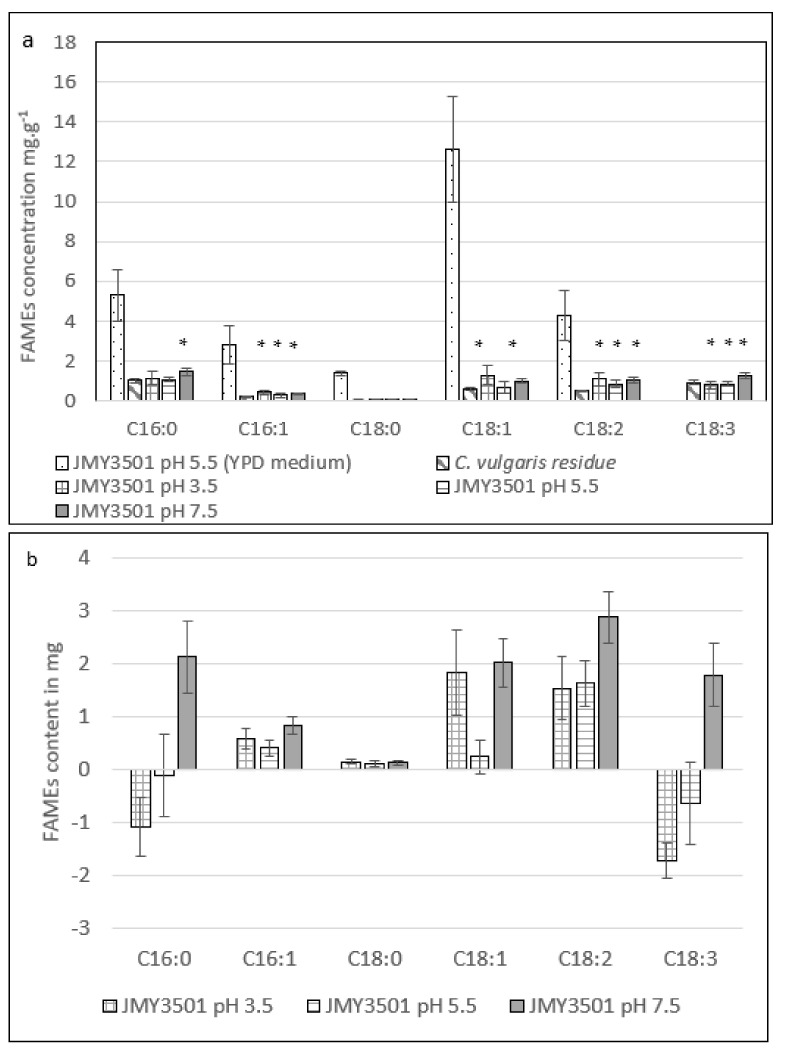
(**a**) Raw data on the fatty acid content of the control (*C. vulgaris* cultivated at a pH of 7.5) and the JMY3501 strain cultivated on YPD at a pH of 5.5 and on *C. vulgaris* residue media for 50 h at different pH levels. (A statistical difference was found between the initial and final concentration of fatty acids in the *C. vulgaris* residue; Kruskal–Wallis test, n = 4, ddl = 1, * significant at *p*-value < 0.05.) (**b**) Evolution of fatty acid quantities in the fermenter after cultivation of the JMY3501 strain in *C. vulgaris* residue for 50 h. These values were obtained by subtracting the initial fatty acid quantities present in *C. vulgaris* residue from the final fatty acid quantities present in the fermenter. (C16:0: palmitic acid; C16:1: palmitoleic acid; C18:0: stearic acid; C18:1: oleic acid; C18:2: linoleic acid; C18:3: α-linolenic acid).

**Table 1 marinedrugs-20-00264-t001:** Biochemical and elementary composition of lipid-extracted *C. vulgaris* residue.

**Biochemical Compound**	**Content (Percentage of Dry Weight)**
Proteins	40.1 ± 5.3
Reducing carbohydrates	15.6 ± 2.1
Fatty acids	0.34 ± 0.02
**Elementary Composition**	**Content (Percentage of Dry Weight)**
Total carbon	34.9 ± 0.4
Total nitrogen	7.8 ± 0.2
Ash	11.2 ± 0.1

**Table 2 marinedrugs-20-00264-t002:** Summary of the growth parameters of *Y. lipolytica* W29 and JMY3501 on YPD (pH 5.5) and on *C. vulgaris* residue media at different pH levels (3.5, 5.5, and 7.5) after 50 h of cultivation.

Strains Medium	pH	Specific Growth Rate (h^−1^)	Cell Concentration (10^6^ Cells·mL^−1^)	Generation Time (h^−1^)	Stationary Phase (h)
*Y. lipolytica* W29 YPD medium	5.5	0.28 ± 0.03	1082.8 ± 4.61	2.50 ± 0.26	35
*Y. lipolytica* W29 Lipid-extracted *C. vulgaris* residue	3.5	0.28 ± 0.05	59.6 ± 8.1	2.56 ± 0.49	15
	5.5	0.27 ± 0.06	200.8 ± 94.2	2.67 ± 0.59	15
	7.5	0.26 ± 0.01	235.4 ± 81.5	2.64 ± 0.09	25
*Y. lipolytica* JMY3501 YPD medium	5.5	0.3238 ± 0.0004	1501.7 ± 33.4	2.14 ± 0.00	35
*Y. lipolytica* JMY3501 Lipid-extracted *C. vulgaris* residue	3.5	0.28 ± 0.08	77.1 ± 13.7	2.60 ± 0.76	15
	5.5	0.28 ± 0.01	62.8 ± 18.3	2.49 ± 0.11	15
	7.5	0.26 ± 0.03	73.9 ± 6.6	2.69 ± 0.30	15

**Table 3 marinedrugs-20-00264-t003:** Total FAMEs and biomass at different pH levels for the W29 and JMY3501 strains cultivated in lipid-extracted *C. vulgaris* residue medium and YPD at a pH of 5.5 after 50 h of cultivation.

Strains	pH	Media	Total Concentration of FAMEs mg.g^−1^	Total Dry Biomass (g)	Total FAMEs (mg)
Lipid-extracted *C. vulgaris* residue	7.5		3.3 ± 0.2	5.4 ± 0.9	17.8 ± 4.1
*Y. lipolytica* W29	5.5	YPD	25.4 ± 18.4	4.0 ± 0.4	101.6 ± 83.8
*Y. lipolytica* W29	3.5	Lipid-extracted *C. vulgaris* residue	4.1 ± 0.5	3.8 ± 0.4	15.6 ± 3.3
*Y. lipolytica* W29	5.5	Lipid-extracted *C. vulgaris* residue	3.9 ± 0.7	4.4 ± 0.3	17.1 ± 4.4
*Y. lipolytica* W29	7.5	Lipid-extracted *C. vulgaris* residue	5.1 ± 0.7	4.0 ± 0.2	20.1 ± 3.9
*Y. lipolytica* JMY3501	5.5	YPD	26.5 ± 6.1	4.4 ± 0.5	115.8 ± 39.8
*Y. lipolytica* JMY3501	3.5	Lipid-extracted *C. vulgaris* residue	4.9 ± 1.5	4.1 ± 0.7	19.1 ± 9.4
*Y. lipolytica* JMY3501	5.5	Lipid-extracted *C. vulgaris* residue	3.9 ± 0.6	5.0 ± 0.1	19.5 ± 3.5
*Y. lipolytica* JMY3501	7.5	Lipid-extracted *C. vulgaris* residue	5.3 ± 0.6	5.26 ± 0.02	27.6 ± 1.2

## Data Availability

Not applicable.

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
