# Peer review of "Evaluation of the Potential of Lipid-Extracted Chlorella vulgaris Residue for Yarrowia lipolytica Growth at Different pH Levels"

_marinedrugs, 2022, doi:10.3390/md20040264_

Round 1

Reviewer 1 Report

The authors describe the use of the delipidated C. vulgaris residue to grow Y. lipolytica wild-type W29 and mutant JMY3501 strains to utilize the microalgae residues or co-products. The manuscript is a bit hard to follow, as some sentences are confusing. I would suggest the authors have a second look and try to rephrase portions of it in the results and discussion sections. Also please keep the tenses in the results section consistent.

Minor comments:

Line 15: Please specify what kind of mutant is JMY3501 strain

Table 1 Please make bold the row 6 (elementary composition)

Figure 1: Please italicize the species name in the legend on each panel and add the pH value of the YPD medium. In the figure legend add color for each shape e.g.: orange diamond. Please add the descriptive marker for YPD medium in point b. Please move a and b to upper left corner of each panel

Line 115 and others: Please italicize species name

Lines 129 – 131: This sentence is confusing please clarify

Line 140 – remove of

Line 149-150: As written the sentence implies that different pH was tested on YPD medium rather than residue medium, please correct

Line 159, 184: Please remove double space

Results section: Please use consistent tense while writing the results section

Line 176: during -> for

Line 190: FAME – please spell out the acronym when using it for the first time

Line 196: Why did authors choose pH 7.5 as a blank control?

Line 197-198 – please spell out the acronym when using it for the first time

Figure 2: Please add in the legend what fatty acids C16:0 and others correspond to so the reader doesn’t have to go back to the text. Why there is such difference between fermenters a and b in fatty acid content?

Figure 3: Please add in the legend what fatty acids C16:0 and others correspond to so the reader doesn’t have to go back to the text. Why there is no fermenters a and b in this case?

Line 275-277: At first you say the that whatever the pH the growth was 3 times lower in the mutant rather than the WT, however that doesn’t hold true for pH 3.5, please rephrase

Line 453: funding-> funded

Author Response

Manuscript Number: Marinedrugs-1632582

Points raised by reviewer 1:

Thank you for your review and pointing out the modifications requested, we have introduced them as described below:

The authors describe the use of the delipidated C. vulgaris residue to grow Y. lipolytica wild-type W29 and mutant JMY3501 strains to utilize the microalgae residues or co-products. The manuscript is a bit hard to follow, as some sentences are confusing. I would suggest the authors have a second look and try to rephrase portions of it in the results and discussion sections. Also, please keep the tenses in the results section consistent.

All the manuscript was checked and improved for the written English. Some parts were modified, changes were tracked in the revised version. The results section has been improved and some parts have been rewritten. Please see the revised version.

Minor comments:

Line 15: Please specify what kind of mutant is JMY3501 strain

Thanks for your valuable comment. The mutations for JMY3501 strain were provided. The sentence in the abstract section was modified as follows: “Both wild-type Yarrowia lipolytica W29 and mutant JMY3501 (which was designed to accumulate more lipids without their remobilization or degradation) showed a similar growth rate of 0.28 h-1 at different pH levels (3.5, 5.5, and 7.5).”. Please see lines from 16 to 19 in the abstract section.

Table 1 Please make bold the row 6 (elementary composition)

Row 6 in Table 1 was modified as requested. Please see the revised version.

Figure 1: Please italicize the species name in the legend on each panel and add the pH value of the YPD medium. In the figure legend add color for each shape e.g.: orange diamond. Please add the descriptive marker for YPD medium in point b. Please move a and b to upper left corner of each panel

Thanks for your comment, modifications were provided in figure 1 as requested. Please see the revised version.

Line 115 and others: Please italicize species name

Thanks for your comment. Species name were Italicized. Please refer to the corrections in the revised manuscript which are tracked.

Lines 129 – 131: This sentence is confusing please clarify

We agreed with this remark, sentences in line 129 were modified as follows: “The final cell concentration of the W29 strain in the C. vulgaris medium at a pH of 5.5 or 7.5 was above 70% of the final cell concentration at a pH of 3.5 (Table 2).” see lines from 197 to 200 in the revised version.

Line 140 – remove of

The word “of” was removed.

Line 149-150: As written the sentence implies that different pH was tested on YPD medium rather than residue medium, please correct

Thanks for your comment, in order to avoid confusion, the sentence from 221 to 223 was modified as follows: “We then examined the effect of pH (3.5, 5.5 and 7.5) on the growth of the JMY3501 strain in the C. vulgaris residue medium compared to that at a pH of 5.5 in the YPD medium.” Please see the revised version.

Line 159, 184: Please remove double space

Thanks for your comment, double spaces were removed as requested.

Results section: Please use consistent tense while writing the results section

This was modified in the results section. Reformulations have been made. Please see the results section in the revised version.

Line 176: during -> for

The paragraph from lines 250 to 254 was removed. The paragraph from lines 255 to 262 has been moved to the discussion section.

Line 190: FAME – please spell out the acronym when using it for the first time

Thanks for your comment. The full term of FAMEs was provided in line 267.

Line 196: Why did authors choose pH 7.5 as a blank control?

pH 7.5 was chosen as a blank control because the homogenization of the microalgal substrate is much better. A sentence was added in line 275 as follows: “We selected this pH because of the improved substrate homogenization”. Please see the revised version.

Line 197-198 – please spell out the acronym when using it for the first time

The full term of GC-FID was provided in line 277 in the revised version.

Figure 2: Please add in the legend what fatty acids C16:0 and others correspond to so the reader doesn’t have to go back to the text. Why there is such difference between fermenters a and b in fatty acid content?

Figure 2 was provided with the legend as requested.   

The difference of fatty acids content in fermenters a and b despite a similar growth, could be explained by the capacity of wild-type to remobilize its lipids during the stationary phase in a rich medium and in presence of a tiny change of physical fermentation conditions like dissolved oxygen.

Figure 3: Please add in the legend what fatty acids C16:0 and others correspond to so the reader doesn’t have to go back to the text. Why there is no fermenters a and b in this case?

Figure 3 was provided as requested.

In this case the mutant was not able to remobilize its fatty acids. Therefore, we didn’t observe a significant difference between the duplicate in the fatty acid concentration.

Line 275-277: At first you say the that whatever the pH the growth was 3 times lower in the mutant rather than the WT, however that doesn’t hold true for pH 3.5, please rephrase

Authors agree with this remark, sentences in line 275 were modified as follows:” Therefore, cell growth in the microalgal medium appeared to be three times lower in the deleted xpr2 gene JMY3501 mutant at pH levels of 5.5 and 7.5 than in the wild-type W29 strain at the same pH levels [40].”. Please see lines from 439 to 441

Line 453: funding-> funded

The word was corrected accordingly.

Reviewer 2 Report

In the subject of lipid production in Y. lipolytica yeast cells much research has been done and many publications have been written. However, the directions of waste management and environmentally friendly technologies for obtaining valuable products should always be promoted. The work presented for review is modest in terms of experimentation. Two strains of Yarrowia lipolytica were used, one natural and one genetically modified obtained by the authors from friendly collections. Only a few batch cultures were performed, yeast growth was determined and the lipdid composition of the biomass was analyzed after 50 h of culture. However, Authors propose an interesting substrate - lipid-extracted Chlorella vulgaris residue without any pretreatment as sole carbon and nitrogen sources to grow of yeasts. From an economic point of view, it is interesting where and how much of this waste is generated annually?

The cultures were conducted in final operating volume of 300 mL - please specify precisely how many ml of sample were taken each time? What volume was ultimately withdrawn from the bioreactor?

An interesting finding of these studies is the fact that during the growth of the JMY3501 strain on the waste substrate at a pH of 7.5, a significant amount of 18: 3 acid was accumulated in cells. Please state what kind of application the authors see for this acid?

Line 153 – Lazar et al. 2014 or [31]?

Author Response

Manuscript Number: Marinedrugs-1632582

Points raised by reviewer 2:

Thank you for your review and pointing out the modifications requested, we have introduced them as described below:

In the subject of lipid production in Y. lipolytica yeast cells much research has been done and many publications have been written. However, the directions of waste management and environmentally friendly technologies for obtaining valuable products should always be promoted. The work presented for review is modest in terms of experimentation. Two strains of Yarrowia lipolytica were used, one natural and one genetically modified obtained by the authors from friendly collections. Only a few batch cultures were performed, yeast growth was determined and the lipid composition of the biomass was analyzed after 50 h of culture. However, Authors propose an interesting substrate - lipid-extracted Chlorella vulgaris residue without any pretreatment as sole carbon and nitrogen sources to grow of yeasts. From an economic point of view, it is interesting where and how much of this waste is generated annually?

Thanks for your question. Sentences have been added in the introduction section as follows: “It has been reported that the global production of microalgae was 15000 tons/year [9]. In 2020, the production of Chlorella vulgaris within European Union was 82 tons [10]. However, there is lack of studies regarding effective utilization of the microalgal residues, which is left after lipid extraction but still enriched with significant amount of proteins, carbohydrates or cellulose, so we aimed to study Chlorella vulgaris residue as substrate [11].” Please see lines from 44 to 49 in the revised version.

The following references has been added in the references section:

  1. Benemann, J. Microalgae for Biofuels and Animal Feeds. Energies 2013, 6, 5869–5886, doi:10.3390/en6115869.
  2. Hossein Hoseinifar, S.; Chew, K.W.; Araújo, R.; Calderón, F.V.; Sánchez López, J.; Azevedo, I.C.; Bruhn, A.; Fluch, S.; Tasende, M.G.; Ghaderiardakani, F.; IImjärv, T.; Laurans, M.; Mac Monagail, M.; Mangini, S.; Peteiro, C.; Rebours, C.; Stefansson, T.; and Ullmann, J. Current Status of the Algae Production Industry in Europe: An Emerging Sector of the Blue Bioeconomy. 2021, doi:10.3389/fmars.2020.626389.
  3. Bhattacharya, M.; Goswami, S. Microalgae – A green multi-product biorefinery for future industrial prospects. Biocatal. Agric. Biotechnol. 2020, 25.

Some reformulations have been made in the methods and Conclusion has been removed.

The cultures were conducted in final operating volume of 300 mL - please specify precisely how many ml of sample were taken each time? What volume was ultimately withdrawn from the bioreactor?

Thanks for your valuable comment. The following sentences were added in the manuscript in the material and methods section “To determine the yeast cell density, 10 mL was taken 12 times and 50 mL was taken once (after 13 hours of growth) from the bioreactor. On average, 10–15 mL of acid or base were added during the culture, so the final bioreactor volume was around 145 mL.” Please see lines from 575 to 577 in the revised manuscript.

An interesting finding of these studies is the fact that during the growth of the JMY3501 strain on the waste substrate at a pH of 7.5, a significant amount of 18: 3 acid was accumulated in cells. Please state what kind of application the authors see for this acid?

Thanks for your comment, as Y. lipolytica is recognized as safe, C18:3 can have several applications such as food, nutraceutical and cosmetics.

Line 153 – Lazar et al. 2014 or [31]?

Thanks for this remark, reference Lazar et al.2014 was removed.

Reviewer 3 Report

In the current manuscript, the authors aim to use biomass from Chlorella vulgaris post-lipid extraction, as nutrient source for Yarrowia lipolytica. Two strains were tested, a wild type and an engineered strain for enhanced lipogenesis. Three different pH regiments were assayed.

The manuscript suffers on several issues. The biological experiments should have been done in triplicates, not duplicates. The variability of FAMEs concentrations (Figure 2) for the duplicate cultures of wild type W29 strain is very large (>2-fold) for any meaningful conclusions to be drawn. Labels in figure 1 Y-axis are incomprehensible; x-axis should be Time (not Times).

Major improvements in the use of English are required. The manuscript is interspersed with expressions that are not used in English and do not convey accurately the intended meaning. There is no flow in the presentation; the words Moreover and Furthermore are repeated.

Author Response

Manuscript Number: Marinedrugs-1632582

Points raised by reviewer 3:

Thank you for your review and pointing out the modifications requested, we have introduced them as described below:

In the current manuscript, the authors aim to use biomass from Chlorella vulgaris post-lipid extraction, as nutrient source for Yarrowia lipolytica. Two strains were tested, a wild type and an engineered strain for enhanced lipogenesis. Three different pH regiments were assayed.

The manuscript suffers on several issues. The biological experiments should have been done in triplicates, not duplicates.

We found that the culture of the wild-type strain on the rich YPD medium shows significant variations in the concentration of fatty acids. With the wild strain, the biological experiments were carried out 4 times, only the two closest results are presented. This phenomenon of variability was not observed when the wild-type was cultivated on C. vulgaris residue. On the other hand, the variations in fatty acids are very low in the case of the mutant strain JMY3501 which is deleted for beta-oxidation, this is the reason why we carried out the experiments only in duplicates.

The variability of FAMEs concentrations (Figure 2) for the duplicate cultures of wild type W29 strain is very large (>2-fold) for any meaningful conclusions to be drawn.

The difference of fatty acids content in fermenters a and b despite a similar growth in YPD medium, could be explained by the capacity of wild-type to remobilize its lipids during the stationary phase in presence of a slight change of physical fermentation conditions like dissolved oxygen. This variability was not observed in chlorella vulgaris residue medium.  

Labels in figure 1 Y-axis are incomprehensible; x-axis should be Time (not Times).

Changes were made accordingly. Please see the revised version.

Major improvements in the use of English are required. The manuscript is interspersed with expressions that are not used in English and do not convey accurately the intended meaning. There is no flow in the presentation; the words Moreover and Furthermore are repeated.

All the manuscript was checked and improved for the written English. Some parts were modified, introduction and results section were re-written and reformulations were done. Please refer to the corrections in the revised manuscript.

Round 2

Reviewer 3 Report

The revised version of the manuscript is much improved.